



# Measurement Report: Variability in the composition of biogenic volatile organic compounds in a southeastern US forest and their role in atmospheric reactivity

Deborah F. McGlynn[1], Laura E. R. Barry[2], Manuel T. Lerdau[2, 3], Sally E. Pusede[2], Gabriel Isaacman-VanWertz[1]

[1]Department of Civil and Environmental Engineering, Virginia Tech, Blacksburg, VA, 24061, USA
[2]Department of Environmental Sciences, University of Virginia, Charlottesville, VA, 22904, USA
[3]Department of Biology, University of Virginia, Charlottesville, VA, 22904, USA

*Correspondence to:* Gabriel Isaacman-VanWertz (ivw@vt.edu)

**Abstract.** Despite the significant contribution of biogenic volatile organic compounds (BVOCs) to organic aerosol formation and ozone production and loss, there are few long-term, year-round, on-going measurements of their concentrations and their impacts on atmospheric reactivity. To address this gap, we present one year of hourly measurements of chemically resolved BVOCs between September 15, 2019 and September 15, 2020, collected at a research tower in Central Virginia in a mixed forest representative of ecosystems in the Southeastern U.S. Concentrations of isoprene, isoprene reaction products, monoterpenes and sesquiterpenes are described and examined for their impact on hydroxy radical (OH), ozone, and nitrate reactivity. Concentrations of isoprene range from negligible in the winter to typical summertime 24-hour averages of 4-6 ppb, while monoterpenes have more stable concentrations in the range of tenths of a ppb up to ~1 ppb year-round. Sesquiterpenes are typically observed at concentrations of <10 ppt, but this represents a lower bound in their abundance. In the growing season, isoprene dominates OH reactivity but is less important for ozone and nitrate reactivity. Monoterpenes are the most important BVOCs for ozone and nitrate reactivity throughout the year and for OH reactivity outside of the growing season. To better understand the impact of this compound class on OH, ozone, and nitrate reactivity, the role of individual monoterpenes is examined. Despite the dominant contribution of α-pinene to total monoterpene mass, the average rate constants for reaction of the monoterpene mixture with atmospheric oxidants is between 20% and 30% faster than α-pinene due to the contribution of more-reactive but less abundant compounds. A majority of reactivity comes from α-pinene and limonene (the most significant low-concentration, high-reactivity isomer), highlighting the importance of both concentration and structure in assessing atmospheric impacts of emissions.

## 1 Introduction

Biogenic volatile organic compounds (BVOCs) are a dominant source of reactive carbon in the atmosphere, with an estimated 90% of BVOCs emitted from natural ecosystems (Guenther et al., 1995, 2012). In the presence of nitrogen oxides ($NO_x \equiv NO + NO_2$), BVOCs can react to form tropospheric ozone ($O_3$), which has deleterious effects on human health and ecosystems (Avnery et al., 2011a, 2011b; Kroll and Seinfeld, 2008; Lim et al., 2012). These reactions also result in the formation of lower volatility gases and secondary organic aerosol (SOA) (Atkinson and Arey, 2003; Guenther et al., 1995; Hallquist et al., 1997; Kroll and Seinfeld, 2008), which have direct and indirect effects on the radiative balance of the atmosphere (The Intergovernmental Panel on Climate Change, 2013). Once emitted, BVOCs react with and destroy $O_3$ or produce $O_3$ through reactions with other oxidants (in particular, the hydroxyl radical) (Kurpius and Goldstein, 2003; Wolfe et al., 2011). The impact of plant emissions on net $O_3$ production versus loss depends on concentrations of $NO_x$, as well as the specific chemistry of the BVOCs emitted (Peake and Sandhu, 1983; Pusede et al., 2014; Trainer et al., 1993). Changes in environmental conditions, pollution levels, phenology and



ecological succession affect plants and ecosystems in ways that change their BVOC emissions and ozone uptake (Faiola et al., 2019; Lerdau et al., 1997; Sadiq et al., 2017; Zheng et al., 2017).

BVOC emissions consist largely of terpenes, including isoprene ($C_5H_8$), monoterpenes ($C_{10}H_{16}$), sesquiterpenes ($C_{15}H_{24}$), and
diterpenes ($C_{20}H_{32}$) (Guenther et al., 2012; Kesselmeier and Staudt, 1999; Laothawornkitkul et al., 2009). These compounds vary widely in their reaction rates with atmospheric oxidants, so the impacts of BVOC emissions on regional atmospheric chemistry and composition vary with atmospheric composition and species composition of the dominant vegetation (Atkinson et al., 1992; Claeys et al., 2004; Geron et al., 2000; Goldstein and Galbally, 2007; Hoffmann et al., 1997; Lee et al., 2006). Compounds that are acyclic or cyclic with endocyclic double bonds tend to react faster with oxidants due to the higher substitution of the alkenyl
carbons in contrast to exocyclic double bonds, which often have one unsubstituted alkenyl carbon (Hatakeyama et al., 1989). Endocyclic monoterpenes (e.g., myrcene, limonene and $\Delta^3$-carene) and sesquiterpenes (e.g., α-humulene and β-caryophyllene) (Atkinson and Arey, 2003; Matsumoto, 2014) also have a greater aerosol formation potential because C-C scission of the double bond is less likely to produce high-volatility fragments (Friedman and Farmer, 2018). Additionally, previous studies that have assessed the reactivity of OH and $O_3$ have found that higher molecular weight BVOCs are emitted at lower rates, but that they
make up an outsize percentage of OH and $O_3$ reactive loss due to their faster reaction rates (Helmig et al., 2006; Holzke et al., 2006; Yee et al., 2018). Therefore, detailed speciated BVOC data are important for understanding reactivity and formation of ozone and SOA. Furthermore, long-term and detailed measurements of BVOCs can assist in mitigating inaccuracies in modeled BVOC emissions and in understanding their contribution in ozone and SOA formation (Porter et al., 2017).

The chemical complexity of BVOCs presents a challenge in understanding both atmospheric oxidant interactions and SOA
production and composition. This problem becomes more complex in a changing climate and, subsequently, with changing ecosystems. For example, emissions have been found to increase during a forest thinning event (Goldstein et al., 2004) and decrease during times of severe drought and elevated $CO_2$ (Demetillo et al., 2019; Holopainen et al., 2018). Additionally, increased herbivory has been shown to increase both total plant emissions as well as the relative proportion of sesquiterpenes, which in turn affects SOA production and composition (Faiola et al., 2018, 2019). Therefore, understanding oxidant budgets, SOA formation,
and future changes to ecosystems and atmospheric composition requires a detailed chemical understanding of BVOCs.

BVOCs have been the subject of a large number of measurements and studies, and an exhaustive overview of available datasets is outside of the scope of this manuscript. Generally, measurement campaigns make tradeoffs between temporal resolution, chemical resolution, and long-term instrument stability. Many campaigns of a few weeks to a few months have provided chemically detailed (i.e., isomer-resolved) measurements of BVOCs with time resolution on the order of hourly (Gilman et al., 2009; Goldstein
et al., 2004; Park et al., 2010; Schade et al., 1999; Schade and Goldstein, 2003), while longer-term (multi-season or multi-year) measurements tend to achieve lower temporal resolution (Guenther et al., 1996; Holdren et al., 1979; Simpson et al., 2012). Some measurements have provided temporal resolution on the order of minutes of seconds by using direct mass spectrometry (e.g., proton transfer reaction mass spectrometry, Davison et al., 2009; Fares et al., 2010; Ghirardo et al., 2010; Greenberg et al., 2003; Kalogridis et al., 2014), but these instruments are unable to resolve isomers that may differ substantially in their reactivity and
physicochemical properties. A few measurement campaigns have collected long-term (many month), temporally resolved (hourly), and chemically detailed (isomer resolved) measurements of a range of BVOCs (Goldstein et al., 2000; Helmig et al., 2016; Kramer et al., 2015; Millet et al., 2005; Panopoulou et al., 2020; Plass-Dülmer et al., 2002; Read et al., 2009; Schade and Goldstein, 2001), but the number of such campaigns is fairly limited and very few are currently ongoing. These long-term, temporally and chemically detailed measurements are important for understanding the impacts and behavior of BVOCs on time scales relevant to atmospheric
processes, from intra-daily to inter-annually. Therefore, to further advance understanding of the role of biogenic emissions with reactions of atmospheric oxidants on timescale of hours to season, we present one year of temporally and chemically resolved





measurements of chemically-resolved BVOCs in a forest canopy that is representative of many ecosystems in the eastern and southeastern U.S., as part of an ongoing measurement site with measurements planned to continue for the indefinite future. We examine here the temporal and seasonal patterns of BVOCs that drive oxidant reactivity. The specific focus of this work is to

understand the extent to which composition of major BVOC classes may vary and how minor but reactive components may drive oxidant chemistry. A major outcome of this work is a detailed characterization of monoterpenes that may allow model descriptions and non-isomer-resolved measurements of this chemical class to more accurately capture its impacts on tropospheric chemistry.

## 2 Methods

### 2.1 Instrument location and operation

In-canopy BVOC concentrations were measured at the Virginia Forest Research Facility (37.9229 °N, 78.2739 °W), located in Fluvanna County, Virginia. The site is located on the east side of the Blue Ridge Mountains and receives some anthropogenic influence from Charlottesville, VA, and surrounding counties, located 25 km to the west of the site. The forest canopy consists predominantly of maple, oak and pine trees and is approximately 24 m tall (Chan et al., In Review). The site houses a 40-meter meteorological tower, with a climate-controlled, internet-connected lab at the bottom that is supplied by line power, known

alternately as "Virginia Forest Research Lab" (VFRL) and "Pace Tower". The measurement period included in this work extends from September 15, 2019 to September 15, 2020, though measurements are ongoing and are anticipated to continue for several years. All results describing seasonality are divided into two separate seasons based on approximate frost dates: the growing season (May-October) and non-growing season (November-April).

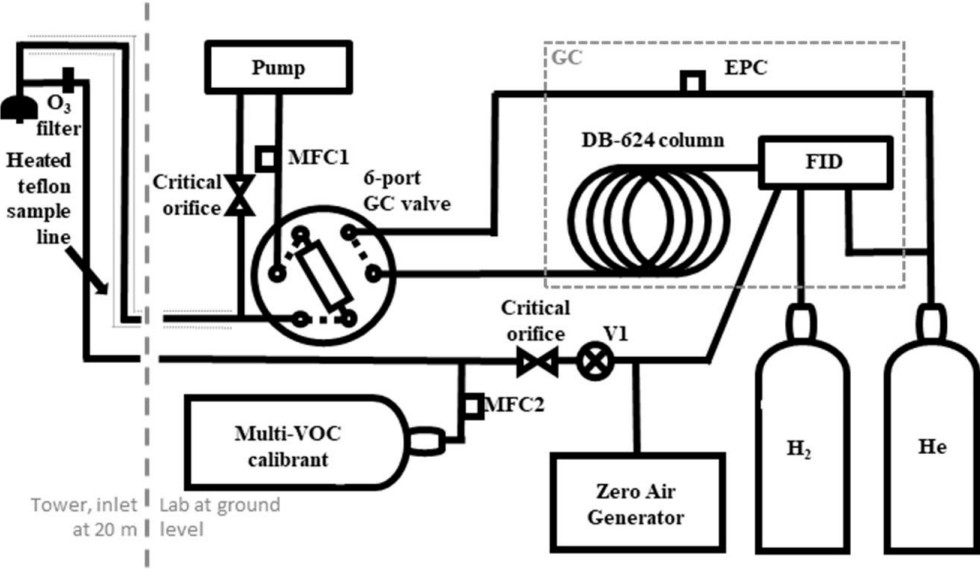

**Figure 1.** A schematic of the VOC-GC-FID set up at the Virginia Forest Research Lab. MFC, mass-flow controller; V1, valve; GC, gas chromatograph; FID, flame ionization detector; EPC, electronic pressure controller; He, helium gas; H₂, hydrogen gas. Small, dashed lines around the inlet, denote the portion that is heated. The large grey dashed line denotes the indicates that the inlet is within the canopy and the rest of the instrument is in the shed, the boxed-in grey dashed line denotes the components that are part of the GC.





Figure 1 depicts the sampling and instrumentation configuration for the automated gas chromatograph-flame ionization (GC-FID) detector used to quantify BVOC concentrations (the "VOC-GC-FID"). Air is pulled from mid-canopy (~20 m above ground level) at 1300 sccm through a 1/4" ID Teflon tube in an insulated waterproof sheath held at 45°C. Ozone is removed from the sample using a sodium thiosulfate infused quartz fiber filter (Pollmann et al., 2005) at the front of the inlet, which is replaced every 4-6 weeks. A subsample of ~70 sccm is concentrated onto a multibed adsorbent trap composed of: (in order of inlet to outlet) 10 mg of Tenax TA, 20 mg of Carbopack X, and 20 mg of Carbopack B, with 15 mg of glass beads between each layer and at the

inlet (from Sigma-Aldrich); prior work (Gentner et al., 2012) used a trap composition and analytical system similar to this instrument to sample compounds between $C_5$ and $C_{14}$. A custom LabVIEW program (National Instruments) operates the instrument for automated sample collection and analysis, with sample collected for 54.5 minutes of each hour (total sample volume: 3.8 L). Following sample collection, the trap is thermally desorbed under a helium backflush at 140±10 °C for 5.5 minutes, transferring analytes through a heated 6-port valve (150 °C) to the head of the gas chromatography column in a GC oven (Agilent 7890B) with

a helium flow rate of 5.5 sccm. GC analysis begins at the end of thermal desorption and proceeds throughout the subsequent sample collection, enabling hourly sample collection and analysis. Analytes are separated using a mid-polarity GC column (Rtx-624, 60m x 0.32mm x 1.8µm, Restek Inc.) and detected by a flame ionization detector (FID). Sample flow is measured by a mass flow controller (MFC1 in Fig. 1, Alicat Scientific), and GC flow is controlled by an electronic pressure controller (EPC in Fig. 1) on-board the GC. Ultrapure hydrogen and helium are provided from compressed gas cylinders (5.0 grade, AirGas) as FID fuel gas

($H_2$), FID makeup gas (He), and GC carrier gas (He). Air for the FID is generated onsite at 30 psig with a zero-air generator (Series 7000, Environics, Inc.).

## 2.2 Calibration and compound identification

For calibration, the sample inlet is overblown with ~1300 sccm zero air from the zero-air generator, optionally mixed with a multi-component calibrant (Apel-Riemer Environmental Inc.) at one of four different flows (generating four different concentrations of

calibrant mixtures). A calibration sample occurs each 7[th] hour, rotating between zero air only, a calibrant at a fixed "tracking" concentration, and a calibrant at one of three other concentrations. Composition and pure concentration of the multi-component calibrant are provided in Table S1 (in brief: 40.3 ppb isoprene, 4.35-17.60 ppb monoterpenes and sesquiterpenes), and diluted into zero air at flows of 10, 25, 50, and 100 sccm to generate dilutions of approximately 140, 60, 30, and 15 times respectively. Estimated limits of detection for isoprene, isoprene oxidation products, monoterpenes, and sesquiterpenes are 20 ppt, 4.3 ppt, 2.2 ppt, and 2.7

ppt, respectively, estimated as the concentration that would yield a chromatographic peak with a height three times the standard deviation of the noise in the chromatographic baseline. Concentrations reported above these levels have an estimated uncertainty of 15%, primarily driven by uncertainty in chromatographic integration (Isaacman-VanWertz et al., 2017). In most cases, concentrations calculated below these values are either reported as 0.0 ppt (in cases when peaks were too small to be integrated), or reported as calculated, but can be considered to have substantial error.




**Table 1.** Compound identities on example chromatogram

| Compound | Symbol | Retention time (s) | $k_{OH}$ (cm$^3$ molec$^{-1}$ s$^{-1}$) | $k_{O3}$ (cm$^3$ molec$^{-1}$ s$^{-1}$) | $k_{NO3}$ (cm$^3$ molec$^{-1}$ s$^{-1}$) |
|---|---|---|---|---|---|
| Isoprene | I | 308 | 1.01 x 10$^{-10}$, a | 1.27 x 10$^{-17}$, a | 7.00 x 10$^{-13}$, a |
| Methyl Vinyl Ketone | IRP1 | 504 | 5.60 x 10$^{-12}$, b | 1.00 x 10$^{-20}$, c | 4.70 x 10$^{-15}$, d |
| Methacrolein | IRP2 | 597 | 5.00 x 10$^{-11}$, b | 1.00 x 10$^{-20}$, c | 9.60 x 10$^{-16}$, d |
| Thujene | M1 | 1397 | 7.10 x 10$^{-11}$, e | 6.20 x 10$^{-17}$, e | 5.5 x 10$^{-12}$, f |
| Tricyclene | M2 | 1409 | 2.66 x 10$^{-12}$, g | 0 | 0 |
| α-pinene | M3 | 1423 | 5.37 x 10$^{-11}$, a | 9.00 x 10$^{-17}$, a | 6.20 x 10$^{-12}$, k |
| α-fenchene | M4 | 1463 | 5.70 x 10$^{-11}$, c | 1.20 x 10$^{-17}$, c | 8.95 x 10$^{-13}$, c |
| camphene | M5 | 1470 | 5.33 x 10$^{-11}$, g | 9.00 x 10$^{-19}$, e | 6.20 x 10$^{-12}$, e |
| sabinene | M6 | 1507 | 1.17 x 10$^{-10}$, g | 8.30 x 10$^{-17}$, g | 1.00 x 10$^{-11}$, k |
| β-pinene | M7 | 1522 | 7.89 x 10$^{-11}$, g | 2.10 x 10$^{-17}$, g | 2.50 x 10$^{-12}$, k |
| cymene | M8 | 1608 | 1.51 x 10$^{-11}$, h | 0 | 9.90 x 10$^{-16}$, h |
| limonene | M9 | 1600 | 1.64 x 10$^{-10}$, g | 6.40 x 10$^{-16}$, g | 1.22 x 10$^{-11}$, k |
| β-phellandrene | M10 | 1620 | 1.68 x 10$^{-10}$, g | 1.80 x 10$^{-16}$, g | 7.96 x 10$^{-12}$, k |
| γ-terpinene | M11 | 1648 | 1.77 x 10$^{-10}$, g | 1.40 x 10$^{-16}$, g | 2.90 x 10$^{-11}$, k |
| α-cedrene | S1 | 2257 | 6.70 x 10$^{-11}$, g | 2.78 x 10$^{-17}$, i | 8.20 x 10$^{-12}$, j |
| β-cedrene | S2 | 2279 | 6.24 x 10$^{-11}$, i | 1.20 x 10$^{-17}$, i | 3.55 x 10$^{-13}$, j |

[a] Atkinson et al. (2006), [b] Paulot et al. (2009), [c] Atkinson et al. (1990), [d] Kerdouci et al. (2010), [e] Pinto et al. (2007), [f] Pfrang et al. (2006), [g] Atkinson and Arey (2003), [h] Corchnoy and Atkinson (1990), [i] Shu and Atkinson (1994), [j] Estimated using King et al. (1999), [k] U.S. Environmental Protection Agency (2012)


While an FID provides nearly-universal quantification of analytes as a function of their carbon content (Faiola et al., 2012; Scanlon and Willis, 1985), it does not provide any chemical resolution. To identify analytes in the samples, a mass spectrometer (MS, Agilent 5977) was deployed in October 2019 and September 2020 in parallel with the FID. Retention times of analytes detected by the two detectors were aligned using the retention time of known analytes (e.g., calibrants). Analytes were identified

by mass spectral matching with the 2011 NIST MS Library and reported retention indices (National Institute for Standards and Technology, 2019). All analytes reported in this work matched the identified compound within the range of reported retention indices and with a cosine similarity of at least 0.85, which previous work has shown indicates a high probability of correct identification (Worton et al., 2017). Data were analyzed using the freely-available TERN software packaged (Isaacman-VanWertz et al., 2017) within the Igor Pro 8 programming environment (Wavemetrics, Inc.).

**2.3 Atmospheric oxidant reactivity and reaction rate calculations**

Reactivity of an individual BVOC and/or a BVOC class to the hydroxyl radical (OHR), ozone (O$_3$R) and nitrate (NO$_3$R) and is calculated as the sum of the products of the concentration and oxidation reaction rate of each BVOC, $i$:

$$OxR_{tot} \ (s^{-1}) = \sum\left(k_{O_x+BVOC_i}[BVOC_i]\right) \tag{1}$$

Published rate constants (units: cm$^3$ molec$^{-1}$ s$^{-1}$) are used where available (Atkinson et al., 1990, 2006; Atkinson and Arey, 2003;

Corchnoy and Atkinson, 1990; Kerdouci et al., 2010; King et al., 1999; Pfrang et al., 2006; Pinto et al., 2007; Pratt et al., 2012; Shu and Atkinson, 1994; U.S. Environmental Protection Agency, 2012) and were otherwise calculated from structure-activity relationships using the Kwok and Atkinson structure activity relationships as implemented by the Estimation Program Interface provided by the U.S. Environmental Protection Agency (King et al., 1999; Kwok and Atkinson, 1995; U.S. Environmental Protection Agency, 2012).





## 3 Results and Discussion

### 3.1 Temporal trends in BVOC concentrations

Observed and quantified BVOCs include isoprene, two isoprene oxidation products methyl vinyl ketone and methacrolein, eleven monoterpene species, and two sesquiterpene species. A sample chromatogram is show in Fig. 2. The detected monoterpene species include α-pinene, β-pinene, β-phellandrene, camphene, limonene, tricyclene, α-fenchene, thujene, cymene, sabinene, and γ-terpinene (Table 1). The sesquiterpene species regularly detected include α-cedrene and β-cedrene, but these are generally present at very low concentrations. Consequently, we expect that not all sesquiterpenes are captured by this instrument and caution that all reported concentrations of sesquiterpenes represent lower bounds.

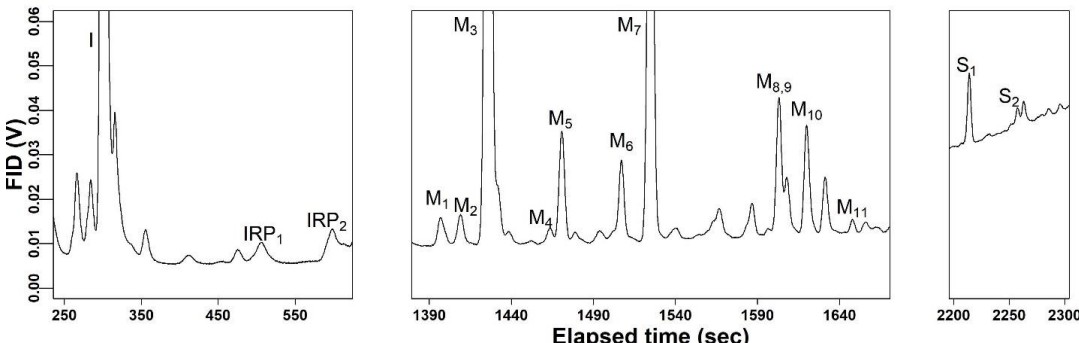

**Figure 2.** A typical GC chromatogram of sampled ambient air collected at the site. The compounds identified on the figure show the range of species found by the instrumental methods. These include isoprene (I), isoprene reaction products (IRP), monoterpenes (M), and sesquiterpenes (S).

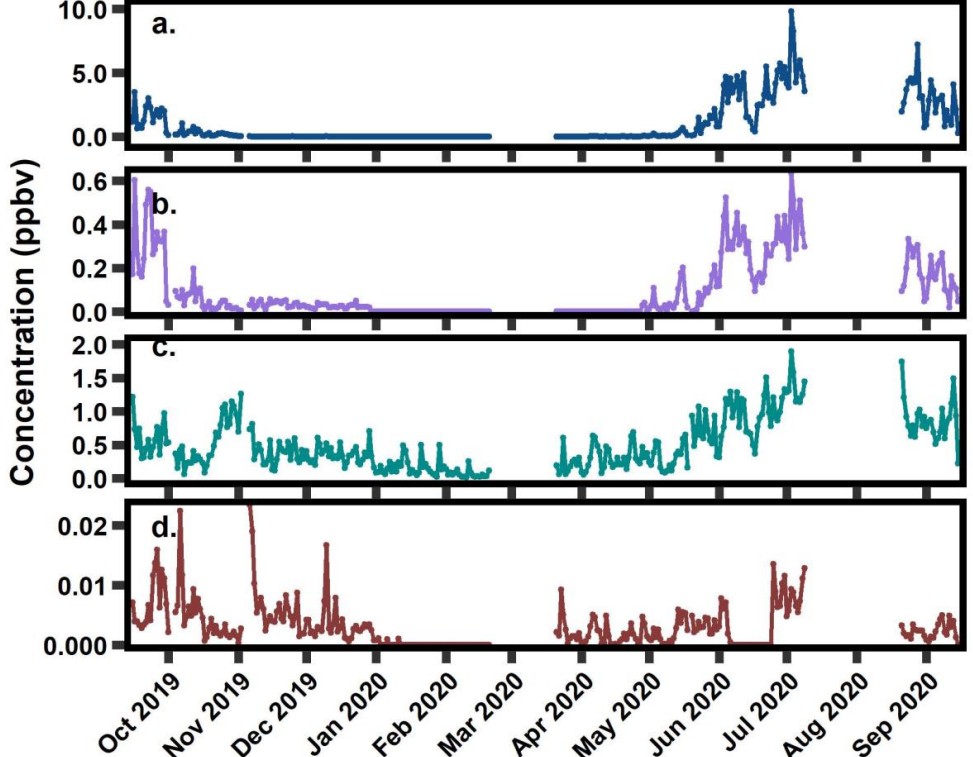

**Figure 3.** 24-hour average concentrations of (a) isoprene , (b) isoprene reaction products (methyl vinyl ketone and methacrolein), (c) monoterpenes, and (d) sesquiterpenes between September 15th, 2019 and September 15th, 2020



Daily 24-hour averaged concentrations of each BVOC class for the measurement period are shown in Fig. 3. Isoprene and its oxidation products (Fig. 3a-b) were near or below detection limits from mid to late-October through early May. Both classes

reached their seasonal peak in late July, with an average hourly isoprene concentration in the growing season of 2.13 (± 2.97) ppb, and near levels of detection in the non-growing season; uncertainties here and throughout represent standard deviations from the mean. The average concentration of summed isoprene oxidation products was 0.27 (± 0.28) ppb in the growing season and 0.02 (± 0.03) ppb in the non-growing season.  The average concentration of summed monoterpenes in the growing season is 0.53 (± 0.62) ppb and 0.20 (± 0.29) ppb in the non-growing season (Fig. 3c). Monoterpenes exhibit a similar high period during the growing

season but are present throughout the non-growing season as well. The average of summer sesquiterpene concentrations in the growing season was 0.01 (± 0.02) ppb and 0.01 (± 0.01) ppb in the non-growing season. Sesquiterpenes, much like monoterpenes, are detected in both the growing and non-growing season. Average concentrations all classes for each season are provided in Table 2.

**Table 2.** Average concentrations, OH (OHR), ozone ($O_3R$), and nitrate ($NO_3R$) reactivities in the growing and non-growing seasons

| | Growing Season Average | | | |
|---|---|---|---|---|
| | Concentration (ppb) | OHR ($s^{-1}$) | $O_3R$ (x $10^{-6}$ $s^{-1}$) | $NO_3R$ ($s^{-1}$) |
| Isoprene | 2.13 ± 2.97 | 5.53 ± 5.30 | 0.78 ± 0.75 | 0.04 ± 0.04 |
| MVK + MACR | 0.27 ± 0.28 | 0.11 ± 0.11 | 0.00 ± 0.00 | 0.00 ± 0.00 |
| Monoterpenes | 0.53 ± 0.62 | 1.95 ± 1.51 | 3.40 ± 2.75 | 0.15 ± 0.11 |
| Sesquiterpenes | 0.01 ± 0.02 | 0.01 ± 0.02 | 0.00 ± 0.00 | 0.00 ± 0.00 |
| | Non-Growing Season Average | | | |
| | Concentration (ppb) | OHR ($s^{-1}$) | $O_3R$ (x $10^{-6}$ $s^{-1}$) | $NO_3R$ ($s^{-1}$) |
| Isoprene | 0.00 ± 0.01 | 0.01 ± 0.02 | 0.00 ± 0.00 | 0.00 ± 0.00 |
| MVK + MACR | 0.02 ± 0.03 | 0.01 ± 0.01 | 0.00 ± 0.00 | 0.00 ± 0.00 |
| Monoterpenes | 0.20 ± 0.29 | 0.70 ± 0.63 | 1.14 ± 1.03 | 0.06 ± 0.05 |
| Sesquiterpenes | 0.01 ± 0.01 | 0.01 ± 0.01 | 0.00 ± 0.00 | 0.00 ± 0.00 |

Diurnal trends in concentrations during the growing season (May-October) and non-growing season (November-April) are shown in Fig. 4 for isoprene, summed isoprene reaction products, summed monoterpenes, and summed sesquiterpenes (α- and β-

cedrene). All terpene classes exhibited the highest concentrations in the growing season (May-October). Isoprene and isoprene





reaction products peak in late afternoon hours, as expected due to the light dependence of isoprene (Guenther, 1997; Lamb et al., 1987; Zimmerman, 1979). Isoprene and reaction product concentrations were typically below the limits of detection in the non-growing season, with little clear diurnal pattern.

In contrast to isoprene, monoterpenes exhibit peak values in the evening hours, which is consistent with previously reported
findings (Davison et al., 2009; Panopoulou et al., 2020). Evening peak values were higher in the growing season than in the non-growing season. Additionally, daytime lows lasted for longer periods of time in the growing season than in the non-growing season due to longer daylight hours driving more photolytic reactions with OH radical and shorter-lasting nighttime boundary layers (Davison et al., 2009). Hourly monoterpene concentrations ranged between 0.10 ppb and 2.94 ppb throughout the year, with the lowest values occurring in the non-growing season (Fig. 3b).

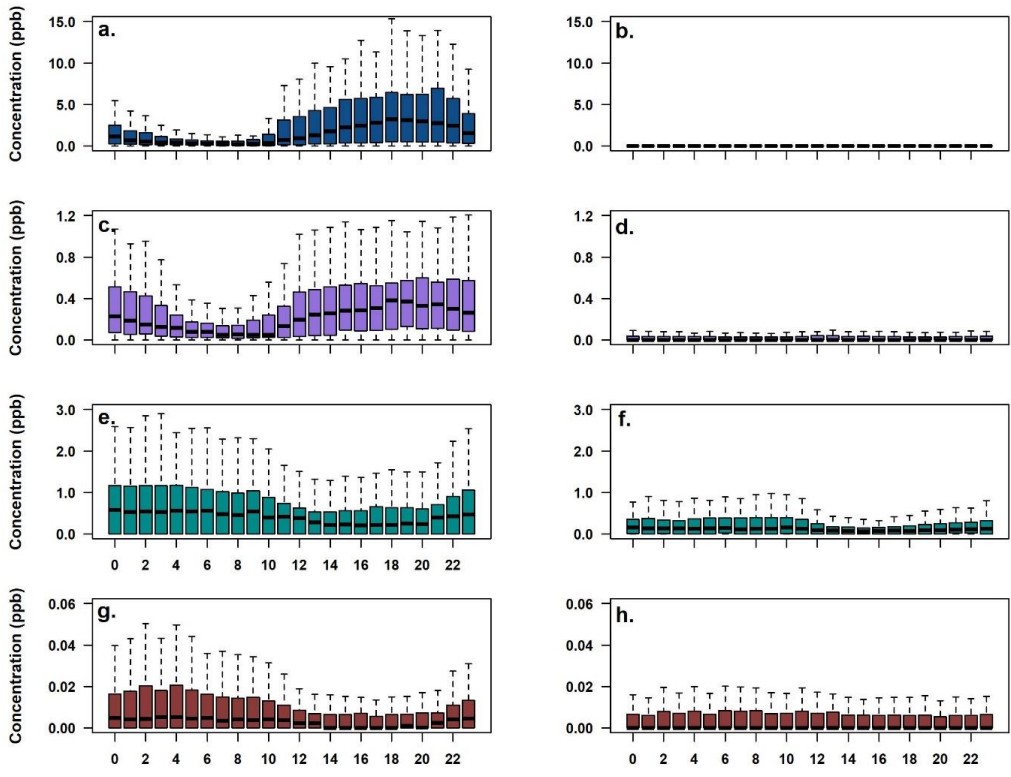

**Figure 4.** Binned hourly boxplots for the four BVOC classes, divided into (left) the growing season, May-October and (right) the non-growing season, November-April. Classes shown are: (a-b) isoprene, (c-d) isoprene reaction products, (e-f) monoterpenes, and (g-h) sesquiterpenes. The plots show the median value as a horizontal line, the bottom and top of each box indicates the 25th and 75th percentiles while the whisker represent 1.5 times the interquartile range. Each box represents the data for each hour of the day.

Sesquiterpene concentrations also exhibited peak diurnal concentrations in the evening in the growing season. Summed concentrations of sesquiterpenes (Fig. 3g and 3h) include only two species, and so represent a lower bound of possible total sesquiterpene concentrations. However, the limit of detection for sesquiterpenes is estimated as 2.7 ppt, so other sesquiterpenes are unlikely to be present at concentrations significantly higher than this. Measured sesquiterpenes therefore provide some insight into the total concentrations of sesquiterpenes. Mean sesquiterpene (i.e., sum of α- and β-cedrene) values were around 0.01 (±
0.02) ppb in the growing season and exhibited similar diurnal variability to monoterpenes. Outside the growing season, the mean sesquiterpene concentration was 0.01 (± 0.01) ppb with little discernable diurnal variability.





### 3.2 Reactivity with atmospheric oxidants

### 3.2.1 OH reactivity

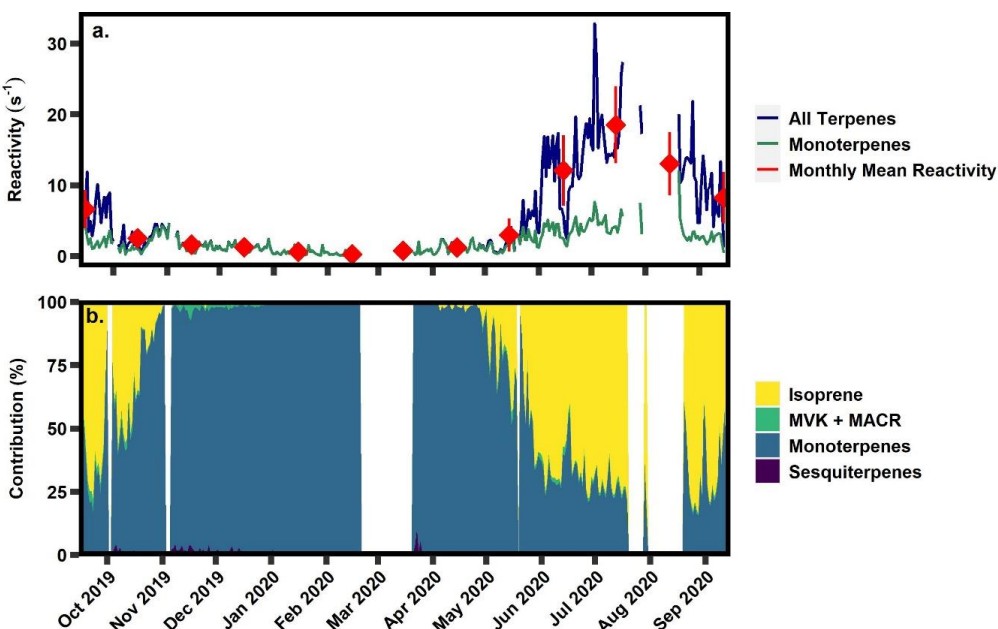

**Figure 5.** (a) Timeseries of 24-hour averaged total OH reactivity of all terpene classes and the monoterpene class, as well as the monthly mean of total OH reactivity. (b) Relative contribution of each of the BVOC classes to OH reactivity.

OH reactivity of total observed terpenes varies seasonally, with a 24-hour average summertime peak of ~32 s$^{-1}$ and growing season

average of 7.60 (± 6.94) s$^{-1}$ driven by isoprene (Fig. 5a). Comparatively, the non-growing season OH reactivity average was 0.73 (±0.67) s$^{-1}$. Reactivity of monoterpenes has weaker seasonality with higher values occurring in the growing season, peaking at 5.00 s$^{-1}$. These values are roughly within the range of previously reported summertime OH reactivity of 1-21 s$^{-1}$ where measurements were taken below ponderosa and coniferous forest canopies and within the canopy of a coniferous forest (Nakashima et al., 2014; Ramasamy et al., 2016; Sinha et al., 2010), though at the higher end, likely due to the measurements in this work occurring directly

within the canopy. While isoprene dominates reactions with OH when present (Fig. 5b), concentrations of isoprene and detected sesquiterpenes are negligible in the non-growing season, causing a steep decline in reactivity. Due to the year-round presence of monoterpenes, these compounds become the dominant source of OH reactivity in the non-growing season. Generally, monoterpenes contribute ~100 % in the non-growing season and ~20-40 % of terpene reactivity in the growing season, with isoprene dominating the balance. Detected isoprene reaction products and sesquiterpenes contribute, on average, <5% to OH

reactivity. While some sesquiterpenes may be below level of detection, sesquiterpenes do not generally have OH reaction rates substantially higher than other more dominant terpenes (Lee et al., 2006) and so are not likely to contribute substantially to OH reactivity.





**3.2.2 Ozone reactivity**

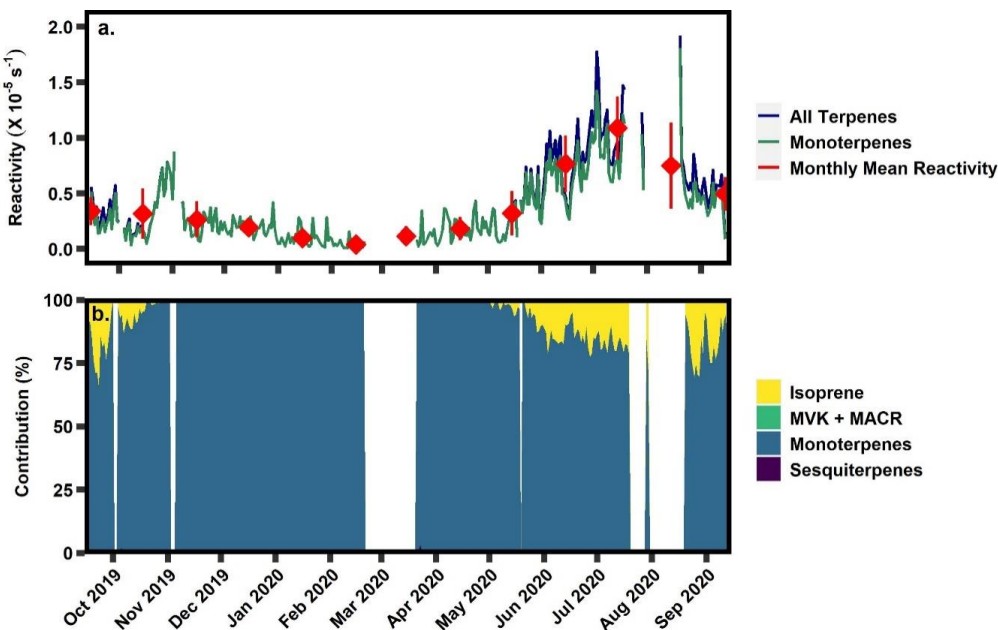

**Figure 6.** (a) Timeseries of 24-hour averaged total ozone reactivity of all terpene classes and the monoterpene class, as well as the monthly mean of total ozone reactivity. (b) Relative contribution of each of the BVOC classes to ozone reactivity.

Total 24-hour averaged $O_3$ reactivity ranges between $0.1\times10^{-6}$ s$^{-1}$ and $1.5\times10^{-6}$ s$^{-1}$ (Fig. 6a) and is almost entirely dominated by

monoterpenes (Fig. 6b), even during the growing season peak (monoterpenes: ~70 %), due to the relatively slow reaction rate of isoprene and its reaction products with ozone. Ozone reactivity decreases in the non-growing season due to both the decline in isoprene, and the decrease in monoterpenes. Average ozone reactivity with isoprene in the growing season is $0.78\times10^{-6}$ ($\pm0.75\times10^{-6}$) s$^{-1}$ while in the non-growing season isoprene does not contribute substantially to ozone reactivity (Table 2). Average ozone reactivity with monoterpenes in the growing season is $3.40\times10^{-6}$ ($\pm2.75\times10^{-6}$) s$^{-1}$ while in the non-growing season it is $1.14\times10^{-6}$

($\pm1.03\times10^{-6}$) s$^{-1}$ The measured isoprene reaction products and sesquiterpenes are not strongly reactive with ozone, and therefore have no significant contribution to ozone reactivity. However, unlike with OH reaction rates, $O_3$ reaction rates of sesquiterpenes are frequently orders of magnitude larger than dominant monoterpenes, so it is possible that low-concentration, highly reactive sesquiterpenes may still contribute non-negligibly to ozone reactivity. Important contributions by low-concentration, high-reactivity sesquiterpenes have been previously shown in other environments, and cannot be excluded by these measurements (Arnts

et al., 2013; Ortega et al., 2007; Wolfe et al., 2011; Yee et al., 2018).





### 3.2.3 Nitrate Reactivity

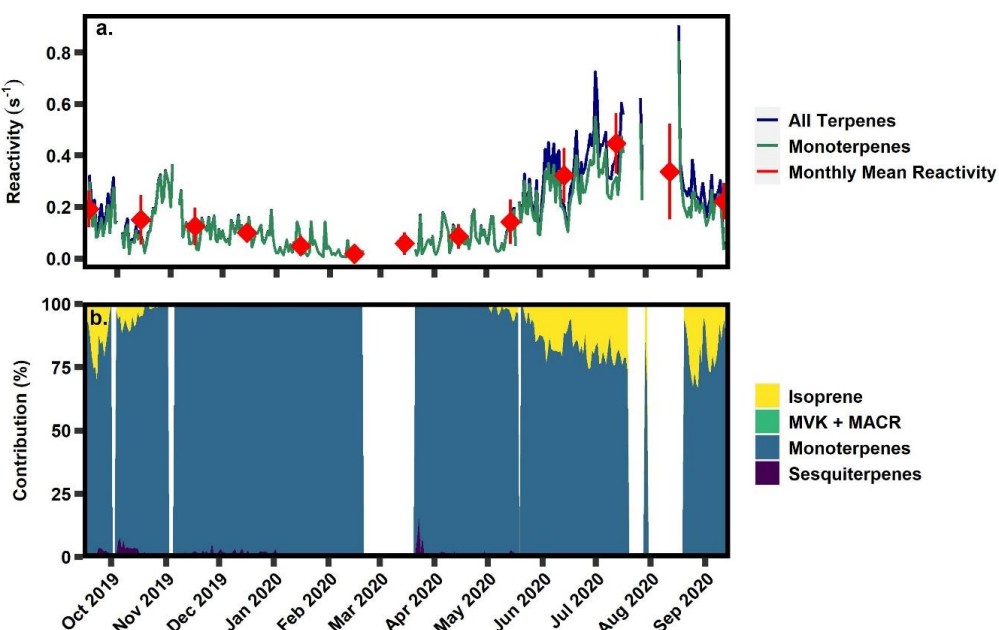

**Figure 7.** (a) Timeseries of 24-hour averaged total nitrate reactivity of all terpene classes and the monoterpene class, as well as the monthly mean of total nitrate reactivity. (b) Relative contribution of each of the BVOC classes to nitrate reactivity.

Total nitrate reactivity of all detected BVOCs is shown in Fig. 7a in addition to the monthly average reactivity and the reactivity of the monoterpene class. The amount that each BVOC class contributes to the total nitrate reactivity is shown in Fig. 7b. Nitrate reactivity of total observed BVOCs varies seasonally, with a summertime peak in 24-average reactivity of ~0.6 s$^{-1}$ driven largely

by monoterpenes. Reactivity of monoterpenes has strong seasonality with higher values occurring in the growing season, peaking at ~0.4-0.6 s$^{-1}$. As in the case of ozone, nitrate reactivity is dominated by monoterpenes due to the slow reaction rates of isoprene and its oxidation products with $NO_3$. Isoprene contributes between 20-40% to nitrate reactivity in the growing season and has a mean hourly average of 0.04 ($\pm$ 0.04) s$^{-1}$. In the non-growing season, isoprene, like isoprene reaction products and sesquiterpenes do not contribute to nitrate reactivity. Monoterpenes dominate nitrate reactivity year-round and have a mean hourly average of

0.15 ($\pm$ 0.11) s$^{-1}$ in the growing season and 0.06 ($\pm$ 0.05) s$^{-1}$ in the non-growing season.

### 3.3 Isomer composition of monoterpenes

Monoterpenes are detected year-round, but small changes in their compositional breakdown (i.e., the relative contribution of different isomers) leads to important changes in their reactivity and chemistry. Total monoterpene 24-hour average concentrations





ranged between 0.10 ppb and 2.00 ppb with the lowest concentrations occurring in the non-growing season daytime and highest

concentrations occurring in the growing season nights (as shown in Fig. 4).

Relative contributions from monoterpene isomers are similar for the highest concentration species between the growing and non-growing seasons (Fig. 8a-b). At nearly all times, α-pinene contributes the most, followed by β-pinene, camphene, limonene, and cymene. OH reactivity (Fig. 8c and 8d) of each isomer roughly follows the distribution of concentrations, driven by the

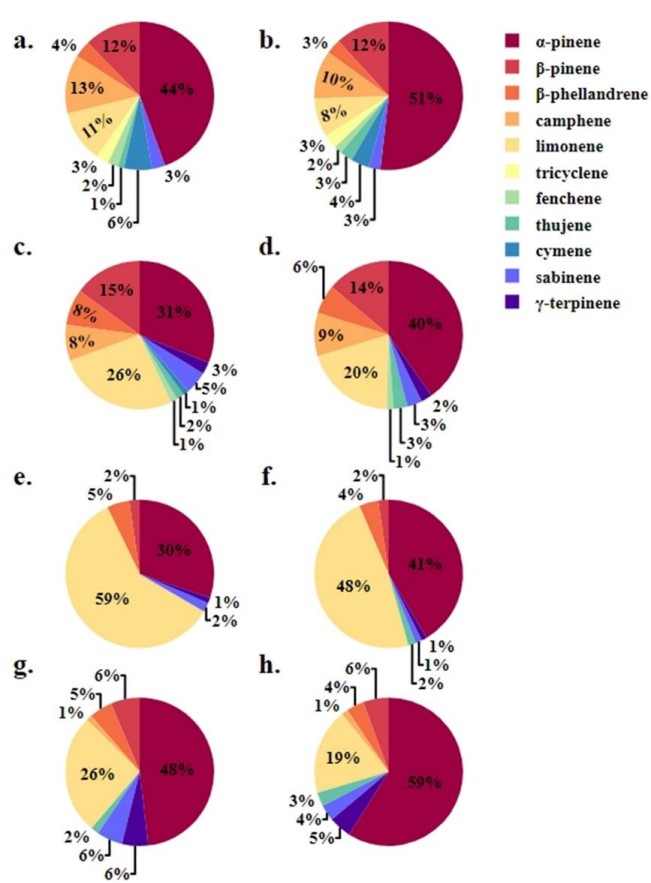

**Figure 8.** A breakdown of by detected monoterpene isomer in the (left) growing and (right) non-growing seasons for (a-b) concentration, (c-d) OH reactivity, (e-f) ozone reactivity, (g-h) nitrate. Values rounded to the nearest percent and values below 1% are not depicted.

relatively narrow range in OH reaction rates for monoterpenes with double bonds (slowest: α-pinene – $5.37 \times 10^{-11}$ cm$^3$·molec$^{-1}$·s$^{-1}$,

fastest: β-phellandrene – $1.68 \times 10^{-10}$ cm$^3$·molec$^{-1}$·s$^{-1}$). There is nevertheless some outsize contribution to OH reactivity by the low-concentration isomers β-phellandrene and limonene, which react quickly due to the presence of multiple double bonds and contribute twice as much to reactivity as they do to concentration. Together, these two species accounts for roughly one-third of OH reactivity, the abundant but less-reactive α-pinene accounts for another ~one-third, and the remaining monoterpenes account for the remainder (mostly β-pinene, camphene, and γ-terpinene). The stability of the concentrations and OH reactivity across a

year of measurements suggests that the observed distribution of isomers is a reasonable average representation of monoterpenes in this ecosystem. While α-pinene is the dominant isomer by far, its lower reaction rate relative to other isomers suggest that it is not





necessarily a good representative proxy species for the broader monoterpene compound class. Instead, a more general quantitative description of the rate at which monoterpenes react with OH (or any other oxidant) would be allow a measurement or estimate of bulk monoterpenes to more accurately be converted into an estimate of their impact on reactivity. Correlations between total
monoterpene oxidant reactivity and total monoterpene concentration (Fig. S1) suggest a bulk monoterpene reaction rate of $k_{OH+MT}$ = $7.0 \times 10^{-11}$ cm$^3 \cdot$molec$^{-1} \cdot$s$^{-1}$, ~30% greater than the reaction rate of α-pinene, and this average rate is relatively temporally stable. It should also be noted that Fig. 8 suggests that OH reaction with faster-reacting, poly-unsaturated, lower-concentration isomers is as likely as reaction with α-pinene.

     The role of structure on atmospheric reactions is even more apparent and critical when considering the reactivity of
monoterpenes with ozone. Despite its relatively low concentration, limonene is the greatest contributor to reactivity with ozone at 59% and 48% in the growing and non-growing season respectively due to an ozone reaction rate 5-10 times faster than that of the more abundant isomers (Fig. 8e-f). Nearly all the rest of ozone reactivity is contributed by the dominant isomer α-pinene (30% and 41%), with a minor contribution from β-phellandrene (5% and 4%), β-pinene (2% and 2%), and γ-terpinene (1% and 1%), while the other isomers are either not reactive with ozone (cymene, tricyclene) or react very slowly with ozone (camphene). Though
the general breakdown of ozone reactivity is qualitatively similar during both the growing and non-growing seasons, there are significant quantitative differences. Due to the greater contribution of limonene in the growing season compared to the non-growing season, the relative importance of limonene compared to α-pinene increases substantially in the growing season, from 1.3:1 to 2.1:1. In other words, reactions of monoterpenes with ozone, at least in this canopy, are dominated by reactions with limonene, with a smaller but significant contribution from α-pinene. The bulk O$_3$ reaction rate with monoterpenes (i.e., the rate that best
converts hourly concentration to reactivity) is $k_{O3+MT}$ = $1.1 \times 10^{-16}$ cm$^3 \cdot$molec$^{-1} \cdot$s$^{-1}$, ~20% faster than α-pinene. However, while this average rate is relatively stable across seasons, there are periods in the growing season during which the average reaction rate of monoterpenes is substantially faster, which could have impacts during these periods (Fig. S1).

     Isomer-dependence of nitrate reactivity is somewhere between O$_3$ and OH, with an outsize impact of limonene, but with a more even split of reactivity across monoterpenes species. These trends may be explained by the reaction behavior of nitrate. Like the
OH radical, nitrate can react with alkenes by either addition to a double bond or abstract a hydrogen, but it has a stronger tendency to add across a double bond, analogous to O$_3$ (Lee et al., 2014; Pfrang et al., 2006). Similar to OH reactivity, limonene contributes an outsized amount to NO$_3$ reactivity in both the growing and non-growing season (26% and 19%). However, for nitrate reactivity, α-pinene remains the dominant component, contributing 48% to reactions with nitrate in the growing season and 59% in the non-growing season. The average reaction rate of α-pinene is also a reasonably good proxy for the bulk reaction rate of monoterpenes
with nitrate radicals, $k_{NO3+MT}$ = ~$6 \times 10^{-12}$ cm$^3 \cdot$molec$^{-1} \cdot$s$^{-1}$.

**4 Conclusion**

Long-term BVOC measurements are imperative for understanding interannual trends in the formation and loss of ozone and SOA, and for improving existing models of BVOC emissions and oxidation. These measurements are difficult, however, without robust measurement techniques that do not require significant maintenance. The use of an automated GC-FID adapted to collect air
samples makes long-term collection of BVOCs in an unmonitored location possible. Using this method, we have collected and are continuing to measure a range of BVOCs in the canopy of a forest representative of the Southeastern U.S., with periodic coupling of a mass spectrometer to allow for identification of the species of interest. The relative ease of this method gives it great potential for additional long-term BVOC monitoring sites to be set up in more locations.





From this study we have gained a greater understanding of the seasonality of BVOCs ranging from isoprene, isoprene reaction products, monoterpenes, and sesquiterpenes. Isoprene is important for OH reactivity, but monoterpenes prevail as the most important BVOC class for ozone and nitrate reactivities. Monoterpenes are observed to be a diverse class of BVOCs with 11 identified compounds detected at the site year-round. While α-pinene is the most dominant species, a few species with lower concentrations but high reactivities (particularly limonene and β-phellandrene) were found to be important contributors to atmospheric reactivity. This finding is most evident for ozone reactivity but is also the case for OH and nitrate reactivity. The distribution of monoterpenes is qualitatively stable throughout the year, though some important quantitative differences are observed. Consequently, the distribution measured here may be a useful description of the "typical" monoterpene chemical class observed in mixed, temperate forests. The bulk reaction rates of the monoterpene class with major atmospheric oxidants presented here therefore provide an improved means to estimate the reactions and impacts of monoterpenes in cases where isomer-resolved measurements are not available (e.g., when measured using direct-air-sampling mass spectrometers (Davison et al., 2009; Ghirardo et al., 2010)).

*Data availability.* The integrated GC-FID BVOC data used in this work is available upon request from the authors.

*Supplement.* The supplement related to this article is available online from the publisher of this article. :

*Author contributions.* DFM conducted the measurement campaign, completed the data analysis, and led the writing of the manuscript. GIVW supervised the study, designed the measurement campaign, directed the data analysis and writing of the manuscript. LERB and SEP was instrumental in the upkeep of the measurement site and provided feedback on the manuscript. MTL provided feedback on the manuscript.

*Competing interests.* The authors declare that they have no conflict of interest.

*Acknowledgements.* This research was funded by the National Science Foundation (AGS 1837882 and AGS 1837891). Tower maintenance and operation was supported in part by the Pace Endowment. D.F.M. and L.E.R.B. were supported in part by Virginia Space Grant Consortium Graduate Research Fellowships. The authors gratefully acknowledge the assistance of Koong Yi, Graham Frazier, and Bradley Sutliff in their support in upkeep and maintenance of the instrument at Pace Tower.

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
