# Peer review of "Measurement Report: Variability in the composition of biogenic volatile organic compounds in a Southeastern US forest and their role in atmospheric reactivity"

_Atmospheric Chemistry and Physics, 2021_

## Author Response (AR1)

The authors would like to thank the reviewers for their comments and feedback on this manuscript. The reviewer comments have improved the quality and clarity of the measurement report. The authors have revised the manuscript in accordance with the reviewer comments. Line numbers referred to in the author response are the updated numbers after revision. The reviewer comments are in **blue** while the author's responses are in **black**:

**Reviewer 1:**

**General comments:**

McGlynn et al. report on long-term chemically resolved BVOC measurements by GC-FID in a temperate forest. Year-round data for chemically speciated BVOCs is still scarce, so the manuscript is a valuable addition to the existing literature. The analysis of the atmospheric impact of the measured terpene species by calculating their OH, NO3 and ozone reactivities adds to the significance of this paper. I recommend the publication of the manuscript in ACP after the following comments are addressed:

We thank the reviewer for their acknowledgement of the contribution of this work to the field. We hope that our revisions adequately address their comments and concerns.

**Specific comments:**

1. 71: In the Goldstein 2000 reference cited here, there are no BVOCs, so it seems misplaced. The Helmig 2016 reference about natural gas VOCs also seems not very relevant in the BVOC context. An important reference to add in the list of long-term chemically resolved BVOC measurements would be https://doi.org/10.5194/acp-18-13839-2018, and maybe, https://doi.org/10.5194/acp-18-3403-2018.

Thank you for those references, they have been added, and the others have been removed.

1. 141 ff: Please comment on the uncertainty of these calculated rate constants. How does the calculation method perform for molecules with known rate constants? And what is the resulting overall uncertainty for your calculated reactivities?

The authors appreciate the reviewer making this point. To address this comment the authors have added this information to the manuscript at line 161:

"Rate constants calculated using structure activity relationships are estimated to be within a factor of  $\sim 2$  of measured rate constants (King et al. (1999)). However, uncertainty of estimated rate constants is not expected to significantly impact calculated reactivity as compounds with the largest contribution to atmospheric reactivity have measured rate constants."

1. 148-152: How complete is the method for monoterpenes? Is this a lower bound, too? E.g. do you know what the peaks between M7 and M8, or between M10 and M11, are?

All identifiable monoterpenes have been labeled and are included in the data and reactivity calculations. Any measurement method is of course a lower bound, as other compounds could be below the limit of detection. However, given the ppt-level limit of detection for monoterpenes, unmeasured isomers are unlikely to contribute significantly. Unlabeled peaks are generally identifiable but are not within the biogenic compound classes focused on in this paper. However, the authors have added a line in the caption of figure 2 to say:

"Unlabeled peaks were not identified to be terpenes or terpene oxidation products and are, in most cases, identifiable as a belonging to a different compound class".

Fig. 5 legend: You did not measure diterpenes which you mention in the introduction, therefore is it fair to call this "all terpenes"?

Diterpenes were mentioned for completeness in referencing other literature but are not detectable with this measurement technique. To address this comment, we have changed the wording in the manuscript to "all measured terpenes".

Fig. 5 caption: The term "total OH reactivity" usually refers to a direct measurement of total OH reactivity. Here you calculated reactivity from measurements of relatively few individual compounds, which means you cannot be sure that you really captured the total. Therefore, please replace the term "total OH reactivity" with "OH reactivity of total observed terpenes" as in the text, or something else, like "calculated OH reactivity".

The authors appreciate this observation and have changed the wording throughout the manuscript to acknowledge that the reactivity presented is calculated.

1. 203-204: It would be good to mention here that these reports are direct measurements of total OH reactivity, so they are not directly comparable to your method. However, these papers usually also include speciated reactivity (e.g. Nakashima et al. does), which you could use to directly compare your terpene reactivity to their terpene reactivity.

We have edited line 239 in the manuscript to say:

"These values are roughly within the range of previously reported direct measurements of summertime OH reactivity of 1-21 s-1 where measurements were taken below ponderosa and coniferous forest canopies and within the canopy of a coniferous forest (Nakashima et al., 2014; Ramasamy et al., 2016; Sinha et al., 2010), though at the higher end, likely due to the measurements in this work occurring directly within the canopy."

**1. 223 f: Please comment on the potential contribution of unmeasured diterpenes.**

The analytical method used to detect biogenic concentrations in this work is not able to detect diterpenes, therefore, we cannot comment on the contribution of diterpenes in this forest. We have made it more clear earlier in the manuscript that we do not have measurements of diterpenes. We note that previous work has generally observed diterpenes at concentrations substantially lower than sesquiterpenes, so their contribution to reactivity is likely comparably

smaller, but we are unable to directly confirm these results with our instrument. We added at line 169:

"Due to the nature of the sample collection, diterpenes many oxygenated species other than MVK and MACR are poorly captured."

1. 235f: What is the sesquiterpene contribution and do you expect an influence of unknown SQTs to NO3 reactivity?

We have addressed this comment by adding at line 274:

"Unmeasured and minimally detected sesquiterpenes are unlikely to contribute substantially to nitrate reactivity as their reaction rates are typically of the same order of magnitude as  $\alpha$ -pinene and isoprene but they are present at concentrations 10-100 times lower (Yee et al., 2018).

Data availability: Using a repository with a doi would be preferable to store the data for the long term and make them more easily accessible to the scientific community. Especially with such long term data this would probably help the modeling community use them.

The data has been made publicly available through Mendeley Data and is citable with an independent DOI. The DOI for the data set is: doi.org/10.17632/jx3vn5xxcn.1

Technical comments:

Caption of Fig. 3, and throughout the manuscript: Sometimes you call MACR and MVK "isoprene reaction products", sometimes "isoprene oxidation products". Please choose one for consistency - I'd suggest "oxidation products", because "reaction products" is more ambiguous.

This has been changed throughout the manuscript to "isoprene oxidation products".

1. 155: Does +/- signify the standard deviation? Please specify by writing "average +/- standard deviation" at least the first time you use it.

**Added.**

Table 2 caption: Average and standard deviations? Please specify.

The caption has been changed to say:

"Average and standard deviation of mixing ratios, OH (OHR), ozone  $(O_3R)$ , and nitrate  $(NO_3R)$  reactivities in the growing and non-growing seasons."

1. 154-162 inconsistent in past and present tense

Fixed.

Caption Fig. 8: the word "reactivity" is missing behind "nitrate"

Added.

1. 253: "would be allow" – remove the "be"

**Addressed.**

**Reviewer 2:**

**General Comments:**

The paper by McGlynn et al. presents a 1-year dataset (Sep-2019 to Sep-2020) of selected biogenic volatile organic compounds (VOCs) from a mixed forest in Central Virgina, S.E. USA. The measurements were performed using an automated gas chromatograph-flame ionization detector (GC-FID). The mixing ratios of isoprene, isoprene oxidation products, monoterpenes and sesquiterpenes were reported and analyzed for their impact on hydroxy radical (OH), ozone, and nitrate reactivity contributions. Summertime average values of isoprene were as high as 6 ppb and had distinct summer max -winter min seasonality, whereas for monoterpenes the mixing ratios generally ranged from few hundred ppt to 1 ppb, throughout the year. A major objective was speciation of monoterpenes to improve model descriptions and non-isomer-resolved measurements of this chemical class to aid tropospheric chemistry studies. This is an interesting and valuable study which would be a great addition to the literature, also

This is an interesting and valuable study which would be a great addition to the literature, also because of the description of the analytical system since BVOCs can be challenging to quantify on hourly temporal scale continuously for a full year. I recommend publication in ACP after the authors have addressed the comments below.

We thank the reviewer for their acknowledgement of the contribution of this work to the field. We hope that our revisions adequately address their comments and concerns.

**Comments:**

Figure 3 and 4 and elsewhere in the text ppb is referred to as concentration. Concentration is always amount of a substance (moles /kg etc..) per unit volume. ppb is nmol per mol and a molar mixing ratio. This should be corrected everywhere in the text and the Figures (e.g. 2 and 3). Figure quality can be improved, please see suggestions below.

This has been changed to mixing ratio throughout the manuscript.

**Abstract:**

**L14**: I suggest replacing isoprene reaction products with isoprene oxidation products here and throughout the manuscript because reaction is more generic.

We appreciate the reviewer pointing out the ambiguity in this wording and have changed it to say isoprene oxidation products throughout the manuscript.

**Introduction:**

L52- mitigating inaccuracies? Perhaps reducing inaccuracies is better choice?

This has been changed to "reducing" inaccuracies.

**Methods and location:**

It is mentioned that the site received air masses with anthropogenic influence and also that the year was classified simply into two seasons namely growing and non-growing season. As the forest is mixed and has both isoprene and monoterpene emitters, it would be useful to some quantitative information on the tree species composition of the forest. Also here and later while interpreting the data, the authors should highlight the known isoprene and monoterpene emitters. Further there is no information provided on the meteorological conditions such a temperature and rainfall and solar radiation in different months of the year. As biogenic emissions are driven by environmental conditions the authors need to do a better job in describing these and also using it for interpreting the ambient data.

The authors agree this is a useful addition to the manuscript, therefore we have added at line 89:

"Ambient temperature in the winter and spring months of January-April (due to data availability), was  $9.6 \pm 6.7$  °C, and in the summer and fall months was  $24.3 \pm 6.0$  °C (Fig. S1). Downwelling shortwave radiation was lower in the winter and spring months ( $141.4 \pm 229.7$  W m-2) than the summer and fall months (January-April) on average ( $235.6 \pm 305.5$  W m-2) and exhibited lower variability (Fig. S1). The forest canopy consists predominantly of maple, oak, and pine and is approximately 24 m tall (Chan, 2011). Roughly three-quarters of trees in the forest are species that shed their leaves in the fall and winter months. Tree species found at the site range from being predominantly isoprene emitters, such as oak to predominantly monoterpene and sesquiterpene emitters, such as pine (Fuentes et al., 1999). Further information pertaining to tree species at the site can be found in Chan (2011)."

Forest composition is available in the cited thesis and is also currently in review in a peerreviewed journal. The table from the work in review is adapted from a published thesis (Chan, 2011) to include only pertinent information on tree species and relative abundance in the forest. The table can be found below for the reviewer's interest, with discussion in this manuscript limited to only a broad description.

| Name                    | Relative Abundance |
|-------------------------|---------------------------|
| Acer rubrum             | 21.9                      |
| Quercus alba            | 14.29                     |
| Fagus grandifolia       | 11.89                     |
| Pinus virginiana        | 9.57                      |
| Nyssa sylvatica         | 7.56                      |
| Liriodendron tulipifera | 6.07                      |
| Cornus florida          | 5.9                       |
| Carya spp.              | 5.82                      |
| Quercus rubra           | 3.92                      |
| Quercus falcata         | 2.68                      |
| Pinus taeda             | 2.45                      |

| Juniperus virginiana    | 1.82 |
|-------------------------|------|
| Quercus marilandica     | 1.77 |
| Kalmia latifolia        | 1.21 |
| Liquidambar styraciflua | 1.08 |
| Carpinus caroliniana    | 0.72 |
| Sassafras albidum       | 0.47 |
| Quercus prinus          | 0.41 |
| Quercus stellate        | 0.19 |
| Ilex opaca              | 0.19 |
| Platanus occidentalis   | 0.06 |
| Populus deltoides       | 0.03 |
| Castanea pumila         | 0.03 |
| Total                   | 100  |

L95: Please add details concerning the inlet residence time, rain events during deployment and efficiency of ozone scrubber.

We have added at line 105: "The residence time of an air sample in the inlet is about 8 seconds."

The ozone scrubber is modeled after Pollmann et al. (2005). The efficacy of the ozone scrubber was tested prior to field deployment using a multiweek test in which it was exposed to laboratory-generated concentrations of 200 ppb, with inline detection post scrubbing confirming penetrating ozone concentrations below the detection limit of an ozone analyzer (Thermo Fisher Model 49i). For field deployment, several (usually 4) such filters are put into the same housing to ensure there is no breakthrough. Following the first several ozone scrubber replacements, the removed scrubber was tested in the laboratory to confirm its efficacy. A brief comment regarding this has been added to the manuscript:

At line 105 we added: "Efficacy of the ozone scrubber was empirically tested by measuring removal over a multiweek exposure to ozone concentrations several times higher than ambient levels. Efficacy was confirmed following deployment by verifying ozone removal of removed filters.".

**L105: how often over the full year was there a need to replace columns, parts and troubleshoot? This would be helpful for readers those who may be interested in using such system.**

We thank the reviewer for this suggestion and have added some information on routine maintenance at line 122:

"A major advantage of deploying a GC-FID in a field setting is the limited required maintenance. The most frequent maintenance required by the system is the replacement of ozone filters every 4-6 weeks. The system also requires hydrogen and helium gas tanks, which last for roughly 6-8 months (though the former could be generated on-site). GC components (traps, columns) require little to no replacement over the time period reported here under normal

operation. The oil-less vacuum pump used to pull samples suffers somewhat from constant use and overheating in the warmest months of the year and therefore had to be replaced after  $\sim 12$  months."

L124-125: 0.0 ppt is so highly significant. Here and in the Table 2 (0.00??) the authors may wish to correct such unrealistic values by below detection limit etc.. please mention how many such instances and values also (what fraction of the dataset?)

Thank you for this observation, the authors have changed values of 0.00 to LOD where applicable.

**L133: please elaborate what is meant by cosine similarity of 0.85 as these are not routine**

Cosine similarity is the standard comparison metric widely used NIST MS library search program. It is calculated as the dot product of the spectra divided by the magnitude of the spectra and is generally main metric used to determine the similarity between two mass spectra for the purposes of identification. Prior work has shown that cosine similar greater than 0.85 is roughly the cutoff beyond which the unknown spectrum is more-likely-than-not to be equal to the library spectrum. (Stein and Scott, 1994; Worton et al., 2017).

We have edited at line 147 to say:

"All analytes reported in this work matched the identified compound within the range of reported retention indices and with a cosine similarity of at least 0.85. This parameter is the preferred spectral comparison method of the widely-used NIST mass spectral library search program, and previous work has shown that a threshold of 0.85 or greater indicates a high probability of correct identification (Stein and Scott, 1994; Worton et al., 2017)."

**L139 please clarify whether the rate constants were corrected for temperature?**

Rate constants were not corrected for temperature, following the approach of other reports from other measurement campaigns. (e.g., Yee et al. (2018)). We speculate that this approach is common in part because many of the measured reaction rates are only well constrained around 298K (e.g., for  $\alpha$ -pinene + O3, the only available review work cataloged by the NIST Kinetics Database provides no temperature dependence parameters). As noted by Heald et al., (2020) this may somewhat overestimate reactivity and night and in the winter (which reactivity is already much lower), but also allows more direct comparison to previous speciated measurements.

The caption of Table 1 has been edited to say: "Compound identities on an example chromatogram and associated rate constants at 298k for OH, ozone, and nitrate."

**Figure 3 : Monoterpenes should be sum of monoterpenes in caption?**

Added.

Between Jan 2020 and April 2020 almost all are close to zero! Some explanation and additional details are required in terms of LAI and environmental conditions.

Also please clarify: Are gaps due to species being below detection limit or instrument issues? Periods when calibration experiments were carried out should be either provided in a separate Table or highlighted in the Figure. Also please mention whether the sensitivity of the compounds changed during the year-long deployment.

At line 180 we added:

"Periods with gaps are due to instrument issues, periods reported as 0 are below the limit of detection (LOD). Many species approached the LOD in the winter and spring months due to low temperatures and decreased incoming shortwave radiation as compared to the warmest months of the year (Fig. S1)."

**Figure 3: Why are isoprene oxidation products 0.6 ppb in Sep-Oct 2019 for isoprene of 4 ppb and also 0.6 ppb for isoprene of 10 ppb in July 2020? This needs to be clarified.**

This is an interesting point. Sources of MVK and MACR are somewhat more complex than can be explained by a fixed ratio to isoprene. NOx levels and anthropogenic emissions influence the oxidation pathway of isoprene, and vehicles can also directly emit MVK and MACR (Biesenthal and Shepson, 1997; Ling et al., 2019). Consequently, the ratio of MVK+MACR concentrations are expected to have an anthropogenic influence, which may be in part what we are observing here. These species are also expected to have longer atmospheric lifetimes due to their lower reaction rates and are therefore more likely to be transported to the site.

To address this in the manuscript at line 187 we have added:

"Interestingly, the ratio of isoprene oxidation products to isoprene is variable over the course of the measurement campaign. In addition to differences in their oxidation rates, these differences may be due in part to anthropogenic emissions both through the influence of  $NO_x$  on isoprene oxidation pathways and the direct emission of MVK and MACR from vehicles (Biesenthal and Shepson, 1997; Ling et al., 2019)."

Figure 3 and Figure 4: Please add the compounds names in the panel for easy readability.

Added.

L155-160: Are the trees without leaves. discuss which trees are known to be high isoprene emitters..which are MT emitters? discuss the leaf phenology at the site during the year.

We have provided additional information on the types of trees found at the research location at line 89 in the manuscript:

"Ambient temperature in the winter and spring months of January-April (due to data availability), was  $9.6 \pm 6.7$  °C, and in the summer and fall months was  $24.3 \pm 6.0$  °C (Fig. S1). Downwelling shortwave radiation was lower in the winter and spring months ( $141.4 \pm 229.7$  W

 $m^{-2}$ ) than the summer and fall months (January-April) on average (235.6 ± 305.5 W  $m^{-2}$ ) and exhibited lower variability (Fig. S1). The forest canopy consists predominantly of maple, oak, and pine and is approximately 24 m tall (Chan, 2011). Roughly three-quarters of trees in the forest are species that shed their leaves in the fall and winter months. Tree species found at the site range from being predominantly isoprene emitters, such as oak to predominantly monoterpene and sesquiterpene emitters, such as pine (Fuentes et al., 1999). Further information pertaining to tree species at the site can be found in Chan (2011)."

**L157: 0.27 $\pm$ 0.28 .here and in other instances please state interquartile range instead of instead of std dev to indicate ambient variability**

We are not clear on the reviewer's reason for suggesting interquartile range. We feel standard deviation is a reasonable way to communicate ambient variability, and note other work following the same approach (e.g., Panopoulou et al. (2020)). If preferred by the editor, we are happy to switch to IQR.

The Results and Discussion can further benefit through comparison to previous studies at similar latitudes. For example L175 -183 the discussion could benefit from temperature an radiation regimes in which emissions are lower or higher. see for example an analyses in the growing season. Detailed analyses of temperature and radiation regimes associated with highest isoprene emission and formation of photochemically formed compounds (see for e.g. Mishra et al. Emission drivers and variability of ambient isoprene, formaldehyde and acetaldehyde in northwest India during monsoon season, Environmental Pollution, Vol. 267, 115538, 2020) would also provide further mechanistic insights.

The authors have added average hourly temperature and downwelling shortwave radiation figures for January 2020-September 15th, 2020 to the supplemental document (Fig. S1) as these can be beneficial in understanding why emissions are high or low at various periods in the year. Though we agree there is a host of potentially interesting insights into these data. Our goal with this work is make available public/peer-reviewed data and a description of the methods by which they were collected. We are working on several manuscripts to more deeply yield mechanistic insights into the relative importance of difference compounds and compound classes.

For the reviewer's interest, here is the figure added to the supplement. The top panel is ambient temperature and the bottom panel is downwelling shortwave radiation. These data, which were collected at the site are for January 1st, 2020 to September 15th, 2020, which covers the majority of the range of the BVOC data presented here. Data between September 15th, 2019 to December 31st, 2019 were not available.